# Factors that determine dependence in daily activities: A cross-sectional study of family practice non-attenders from Slovenia

Antonija Poplas Susič[1,2☯], Zalika Klemenc-Ketiš[1,2,3☯]*, Rok Blagus[4☯], Nina Ružić Gorenjec[1,4☯]

**1** Ljubljana Community Health Centre, Ljubljana, Slovenia, **2** Department of Family Medicine, Faculty of Medicine, University of Ljubljana, Ljubljana, Slovenia, **3** Department of Family Medicine, Faculty of Medicine, University of Maribor, Maribor, Slovenia, **4** Institute for Biostatistics and Medical Informatics, Faculty of Medicine, University of Ljubljana, Ljubljana, Slovenia

☯ These authors contributed equally to this work.
* zalika.klemenc@um.si

**Data Availability Statement:** Data from this study are available on request. This is due to legal restrictions of our organisation which does not allow to publicly show the database. Contact

## Abstract

### Background

Independence in daily activities is defined as the ability to perform functions related to daily living, i.e. the capacity of living independently in the community with little or no help from others.

### Objective

We focused on non-attenders as a subgroup of patients whose health status is not well known to family practice teams. Our goal was to estimate the prevalence of dependence and its severity level in the daily activities of patients, and to determine the factors that are associated with the occurrence of dependence.

### Design

Cross-sectional observational study.

### Settings and participants

Data was obtained in family medicine settings. Participants in the study were adults living in the community (aged 18 or over) who had not visited their chosen family physician in the last 5 years (non-attenders) and who were able to participate in the study. Through the electronic system, we identified 2,025 non-attenders. Community nurses collected data in the participants' homes. The outcome measure was dependence in daily activities, assessed through eight items: personal hygiene; eating and drinking; mobility; dressing and undressing; urination and defecation; continence; avoiding hazards in the environment; and communication.

### Results

The final sample consisted of 1,999 patients (98.7% response rate). The mean age was 59.9 (range 20 to 99). Dependence in daily activities was determined in 466 or 23.3% (95%

information for data request: Community Health
Centre Ljubljana, Slovenia, email: taj.irroz@zd-lj.si.

**Funding:** This work was supported by the Norway
Grants and the Government Office for Development
and European Cohesian Policy in Slovenia (No.
4300-367/2014). RB was partly supported by the
Slovenian Research Agency (Methodology for Data
Analysis in Medical Sciences, P3-0154). ZKK was
partly supported by the Slovenian Research
Agency (Research in the field of Public Health, P3-
0339).

**Competing interests:** The authors have declared
that no competing interests exist.

CI: [21.5, 25.2]) of the patients. Older patients (over 60 years), with at least one chronic disease, increased risk of falling, moderate feelings of loneliness and a lower self-assessment of health were statistically significantly more likely to be dependent in their daily activities, according to our multivariate model.

## Conclusions

A considerable proportion of family practice non-attenders were found to be dependent in daily activities, though at a low level. We identified several factors associated with this dependence. This could help to identify people at risk of being dependent in daily activities in the general adult population, and enable specific interventions that would improve their health status.

## Introduction

Independence in daily activities is defined as the ability to perform functions related to daily living, i.e. the capacity of living independently in the community with little or no help from others [1]. Basic activities in daily living consist of self-care tasks, including bathing and showering; personal hygiene and grooming (including brushing/combing/styling hair); dressing; toilet hygiene (getting to the toilet, cleaning oneself, and getting back up); functional mobility; and self-feeding (not including cooking or chewing and swallowing) [2]. In the literature, the term disability is also used, defining six disability types: hearing, vision, cognition, mobility, self-care, and independent living [3]. When one or more of these activities is affected, a person is dependent in daily activities.

A study of the prevalence of disability in the adult population of the USA determined that it was about 20–25% [4, 5] higher in older adults and women [4]. Most studies, however, have focused on daily activities in older populations (people over 65 or 75 years old). They found that in older populations, 20–70% of people reported difficulties and/or disabilities in carrying out daily activities [6, 7]. There are different risk factors for becoming dependent in daily activities, such as impaired cognition, depression, comorbidity, low frequency of social contacts, low level of physical activity, poor self-perceived health, hospitalization, being female, feelings of loneliness, smoking, and visual impairment [6, 8, 9].

Many patients with disabilities do not have equal access to healthcare, education and employment opportunities or the services they need, and they are often excluded from everyday activities [10]. Patients that are dependent in daily activities often have unmet needs and expectations [11–13]. Often, they also have multiple and complex health needs [14], requiring an interdisciplinary and multidisciplinary approach to identifying such people [15].

Foregone care is an important patient safety problem. People may miss the appropriate care due to problems with accessibility, to avoid stressful treatments, or because of financial constraints [16]. Family practice non-attenders (people that forgo their required health care) are a specific group of individuals with significantly different characteristics from those of attendees. Foregone care is associated with the male gender [17], younger age [16, 18], married marital status [18], unemployment [18], lower education [17, 18], low socio-economic status [17], greater health needs [16], and perceived economic inadequacy [16]. Additionally, they appear to value their health less strongly and are less likely to believe in the efficacy of health checks [17].

In our study, we were interested in non-attenders as a subgroup of patients whose health status is not well known to a family practice team. Such patients could have limitations in daily activities which prevent them from visiting their family medicine practice. We wanted to

estimate the prevalence and severity of dependence in daily activities in these patients, and to determine the factors that are associated with the occurrence of dependence. This would provide valuable insights that could enable specific interventions in the non-attenders who may have unmet health care needs.

## Materials and methods

### Type of study and settings

This was a cross-sectional observational study in a family medicine setting. It was part of a larger study entitled "Upgraded Comprehensive Patient Care", financed by Norway grants and the Government Office for Development and European Cohesion Policy in Slovenia (No. 4300-367/2014). The larger study focused on the comprehensive management of patients with difficulties in access to primary health care, and took place in the largest Slovenian health care centre, Ljubljana Community Health Centre, which encompasses the capitation of 450,000 people out of the total Slovenian population of 2.1 million. From this study, already two papers have already been published. One focused on the description of the comprehensive model of management of patients and is not based on the data from the study [19]. The other focused on malnutrition and used a different subsample from the study [20]; the aims were different and hence the results were different when compared to this article.

The National Ethics Committee (No. 0120-138/2016-2) approved the study.

### Participants

The participants in the study were adult patients (aged 18 or over) living in the community who had not visited their chosen family physician in the last 5 years (non-attenders) and who were able to participate in the study. Each family medicine practice in the Ljubljana Community Health Centre provided a list of non-attenders derived from the electronic system. This list comprised 2,025 people eligible for inclusion in the study, and they were all invited to participate. The final sample consisted of 1,999 patients (98.7% response rate).

Informed consent was obtained from all patients.

### Data collection

The study commenced in September 2015, and community nurses collected data in the patients' homes for eight months. Before the nurses began to collect the data, they received training in the tools used in the study.

The data were collected from September 2015 until May 2016. The community nurses visited the patients at home and assessed different health aspects using questionnaires on chronic diseases and screening tools already used in upgraded family medicine practices (see 'Outcome measures and other variables'). All the data were anonymously entered into an electronic database and analyzed.

### Outcome measures and other variables

We assessed dependence in daily activities through eight items: personal hygiene; eating and drinking; mobility; dressing and undressing; urination and defecation; continence; avoiding hazards in the environment; and communication. Each item was assessed by a community nurse on a 4-point Likert scale (1—independent, 2—low dependency, 3—high dependency, 4—total dependency). A joint score for dependence in daily activities was computed as an average of the eight items, ranging from 1 to 4 in steps of 0.125, where higher values indicated a higher level of dependence. A patient was considered independent if the score was equal to 1,

and to have low dependency with a score of 1.125–2.0, high dependency with a score of 2.125–3.0, and total dependency with a score of 3.125–4.0. The categories of dependency are similar to the Lawton index of dependency in daily activities [21]. Our primary outcome was dependence in daily activities (dependent: scores ≥1.125 vs. independent: score 1). The secondary outcome was the level of dependence in daily activities in the dependent patients.

The community nurses also collected data on demographic characteristics (age, gender) and health related variables. The community nurses measured weight and height to compute BMI, waist circumference, systolic and diastolic blood pressure, and oxygen saturation, and they felt the peripheral pulses on the patients' legs (present/not present/leg amputation). They asked the patients to self-report the presence of chronic diseases (cardiovascular diseases, diabetes, hypertension, chronic obstructive pulmonary disease (COPD), asthma, depression, osteoporosis, benign prostatic hyperplasia, obesity, colon cancer, breast cancer, neurological diseases, and dementia). If chronic diseases were present, we also noted the number of chronic diseases (of the 13 listed above), which was recorded as '0', '1', '2' and '3 or more'.

BMI was divided into four categories: underweight ($<20.0$ kg/m$^2$), normal (20.0–25.0 kg/m$^2$), overweight (25.1–29.9 kg/m$^2$), and obese ($\geq30.0$ kg/m$^2$) [22]. Waist measurement was divided into normal and risky (for women ≥88 cm, for men ≥102 cm) [23].

Family function was determined by the use of APGAR, measuring five constructs: Adaptation, Partnership, Growth, Affection, and Resolve. The family's APGAR evaluates family functionality regardless of the stage of life of the family members. It provides an objective assessment of family function useful for determining the familial context of the patients. Each of the five constructs is assessed on a three-point scale ranging from 0 (hardly ever) to 2 (almost always) [24]. We categorized family function into poor (APGAR score 0–7) and good (APGAR score 8–10). For the assessment of the risk of malnutrition, we used the Malnutrition Universal Screening Tool (MUST) [25], where a score of one point or more (out of six possible points) was considered as an increased risk of malnutrition. For determining the risk of falling, we used the MORSE fall scale [26] where a score of 25 points or more (out of 125 points) was considered as an increased risk of falling.

We also determined a self-assessment of current health, pain intensity, and feelings of loneliness on a 10-point Likert scale, where a score of 10 represented completely satisfied with current health, the strongest pain, and completely lonely, respectively.

The sample was thoroughly described using all the available variables (Tables 1 and 2) that were measured in the larger study context (see 'Type of study and settings'). To facilitate interpretability, several variables were omitted in the multivariate analysis of the outcome because either: (1) there were more variables describing a similar characteristic where only the clinically most relevant variable was included in the model (e.g. BMI was chosen over waist measurement) [27]; or (2) a variable itself was not clinically relevant for dependence in daily activities (e.g. diastolic and systolic blood pressure, blood oxygen saturation) [6, 8, 28]. Based on this background knowledge, we considered the following nine covariates for the multivariate analysis of dependence in daily activities: gender; BMI; number of chronic diseases (recorded as '0', '1','2', '3 or more'); family function; increased risk of falling; age; self-assessment of current health; pain intensity; and feelings of loneliness.

## Statistical analysis

The categorical variables were summarized by frequencies and percentages, and the numerical variables by medians and interquartile ranges (IQR), presented in Tables 1 and 2. Clopper-Pearson confidence intervals (CI) were calculated for proportions.

**Table 1. Demographic, clinical, and psychosocial characteristics of the sample (categorical variables).**

| Characteristic (Number of non-missing values) | All patients n (%) | Dependent in daily activities n (%) | Independent in daily activities n (%) |
|---|---|---|---|
| **Gender: female** (n = 1999) | 1256 (62.8) | 310 (66.5) | 946 (61.7) |
| **Body mass index** (n = 1970) | | | |
| Underweight | 117 (5.9) | 34 (7.5) | 83 (5.5) |
| Normal | 706 (35.8) | 152 (33.3) | 554 (36.6) |
| Overweight | 701 (35.6) | 158 (34.7) | 543 (35.9) |
| Obese | 446 (22.6) | 112 (24.6) | 334 (22.1) |
| **Risky waist measurement** (n = 1570) | 785 (50.0) | 242 (56.4) | 543 (47.6) |
| **Poor family function** (n = 1974) | 238 (12.1) | 92 (20.2) | 146 (9.6) |
| **Increased risk of malnutrition** (n = 1619) | 211 (13.0) | 67 (16.1) | 144 (12.0) |
| **Increased risk of falling** (n = 1922) | 464 (24.1) | 362 (79.0) | 102 (7.0) |
| **Peripheral pulses on legs** (n = 1473) | | | |
| Present | 1447 (98.2) | 372 (94.2) | 1075 (99.7) |
| Not present | 15 (1.0) | 12 (3.0) | 3 (0.3) |
| Leg amputation | 11 (0.8) | 11 (2.8) | 0 (0.0) |
| **Chronic diseases** (n = 1977) | | | |
| No diseases | 1098 (55.5) | 75 (16.4) | 1023 (67.4) |
| 1 disease | 430 (21.6) | 136 (29.7) | 294 (19.4) |
| 2 diseases | 263 (13.3) | 129 (28.2) | 134 (8.8) |
| 3 or more diseases | 186 (9.4) | 118 (25.8) | 68 (4.5) |
| **Cardiovascular disease** (n = 1993) | 246 (12.3) | 168 (36.2) | 78 (5.1) |
| **Diabetes** (n = 1997) | 213 (10.7) | 93 (20.0) | 120 (7.8) |
| **Hypertension** (n = 1996) | 624 (31.3) | 287 (61.7) | 337 (22.0) |
| **COPD** (n = 1997) | 49 (2.5) | 30 (6.5) | 19 (1.2) |
| **Asthma** (n = 1997) | 58 (2.9) | 28 (6.0) | 30 (2.0) |
| **Depression** (n = 1997) | 84 (4.2) | 35 (7.5) | 49 (3.2) |
| **Osteoporosis** (n = 1996) | 88 (4.4) | 59 (12.7) | 29 (1.9) |
| **Benign prostatic hyperplasia** (n = 729) | 32 (4.4) | 13 (8.7) | 19 (3.3) |
| **Obesity** (n = 1997) | 114 (5.7) | 40 (8.6) | 74 (4.8) |
| **Colon cancer** (n = 1997) | 17 (0.9) | 7 (1.5) | 10 (0.7) |
| **Breast cancer** (n = 1254) | 24 (1.9) | 10 (3.2) | 14 (1.5) |
| **Neurological diseases (excl. dementia)** (n = 1999) | 19 (1.0) | 18 (3.9) | 1 (0.1) |
| **Dementia** (n = 1999) | 21 (1.1) | 21 (4.5) | 0 (0.0) |

Categorical variables (number (n) and percentage (%) of all patients and in each group separately; COPD = chronic obstructive pulmonary disease).

The multivariate logistic model was fitted for dependence in daily activities (dependent vs independent) using all nine of the covariates listed above, where the non-linear effects for numerical variables were included, if required, according to goodness-of-fit analysis (conducted using the tests based on the cumulative sums of the ordered residuals). We used a sample size of 1814 patients who had no missing values for all 9 covariates, of which there were 423 events. This met the requirement for a sufficient number of events per variable, as in our final multivariate logistic model 16 coefficients were estimated. The omitted 185 patients with missing values are comparable to the used subsample of 1814 patients in terms of the outcome and demographic characteristics (see S1 Table). The discriminative ability of the logistic model was estimated by means of a ROC (receiver operating characteristic curve) analysis reporting the area under the ROC curve (AUC), pseudo $R^2$, and their bias-corrected versions computed through resampling validation of the model using 1000 bootstrap repetitions.

**Table 2. Demographic, clinical, and psychosocial characteristics of the sample (numerical variables).**

| Characteristic (Number of non-missing values) | All patients Median (IQR) | Dependent in daily activities Median (IQR) | Independent in daily activities Median (IQR) |
|---|---|---|---|
| **Age** (years) (n = 1999) | 61.0 (41.7–76.0) | 81.2 (69.8–87.5) | 53.6 (38.5–67.1) |
| **Systolic blood pressure** (mmHg) (n = 1983) | 130 (120–140) | 130 (120–140) | 125 (115–137) |
| **Diastolic blood pressure** (mmHg) (n = 1983) | 75 (70–80) | 75 (70–80) | 75 (70–80) |
| **Blood oxygen saturation** (%) (n = 1291) | 97.0 (96.0–98.0) | 97.0 (96.0–98.0) | 97.0 (96.6–98.0) |
| **Self-assessment of current health** (Scale 1–10) (n = 1984) | 8 (5–9) | 5 (4–7) | 8 (6–9) |
| **Pain intensity** (Scale 1–10) (n = 1971) | 1 (1–3) | 3 (2–5) | 1 (1–2) |
| **Feelings of loneliness** (Scale 1–10) (n = 1971) | 1 (1–3) | 3 (1–5) | 1 (1–2) |

Numerical variables (median and interquartile range (IQR) for all patients and each group separately).

A *p* value of less than 0.05 was considered statistically significant. All analyses were carried out with R statistical software, version 3.4.3 [29]; the package gof [30] was used for goodness-of-fit analysis; and the package rms [31] was used for bias-corrected AUC and pseudo $R^2$.

## Results

### Sample description

The sample consisted of 1,999 patients (98.7% response rate). The mean age was 59.9 (range 20 to 99 years), and 62.8% were women. Dependence in daily activities was determined in 466 or 23.3% (95% CI: [21.5, 25.2]) of the patients, but only 5.0% of the patients had a dependence score higher than 2. The whole distribution of the level of dependence in daily activities in dependent participants is shown in Fig 1. All the demographic, clinical and psychosocial characteristics of the sample are presented in Tables 1 (for categorical variables) and 2 (for numerical variables). There were 58.2% overweight or obese patients (50% had a risky waist measurement); 44.5% had at least one chronic disease.

### The presence of dependence in daily activities

The multivariate model for dependence in daily activities (dependent vs. independent) (Table 3) discriminated the data very well with AUC = 0.955 (bias-corrected 0.951), and showed a statistically significant association of dependence in daily activities with age (*p*<0.001); chronic diseases (*p*<0.001); increased risk of falling (*p*<0.001, OR 16.9 with 95% CI [11.6, 24.6]); feelings of loneliness (*p* = 0.001); and self-assessment of current health (*p*<0.001). Gender, BMI, family function and pain intensity were not significantly associated with dependence in daily activities. Patients with chronic diseases had a statistically significantly higher risk of dependence in daily activities than those without chronic diseases (one vs. no disease: *p* = 0.003, OR 2.1 with 95% CI [1.3, 3.5]), whereas the actual number of chronic diseases did not appear to be important. In Fig 2, variables with quadratic effect (age, feelings of loneliness and self-assessment of current health) are presented graphically. For age, a positive effect was observed after the age of about 60. Feelings of loneliness had a positive effect up to a score of about 5, while patients with extremely deep feelings of loneliness had low risk of dependence in daily activities. With regard to the self-assessment of current health, there was no difference in the risk of dependence up to a score of about five. Afterwards, better self-assessment decreased the probability of dependence.

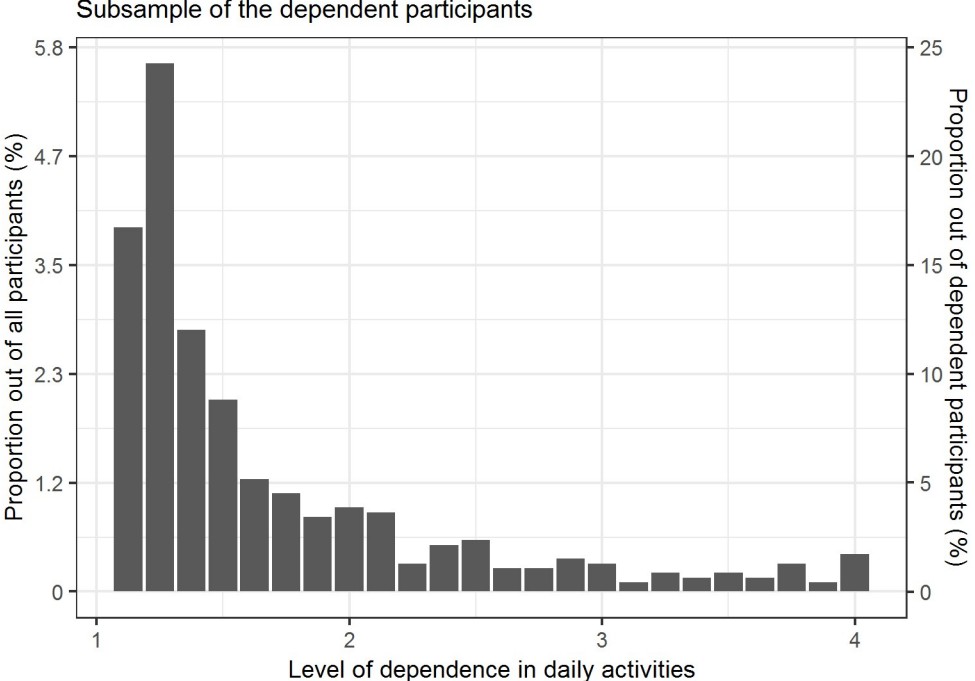

**Fig 1. The distribution of the level of dependence in daily activities in dependent participants.**

## Discussion

Our study showed that almost a quarter of family practice non-attenders were dependent in daily activities, but most of them had only low levels of dependence in daily activities. Patients with increasing age (after the age of 60), at least one chronic disease, increased risk of falling, moderate feelings of loneliness and lower self-assessment of health were statistically significantly more likely to be dependent in their daily activities, according to our multivariate model.

Most of the dependent participants had only a low level of dependence, with a dependence score higher than 2 (on a scale from 1 to 4) in only 5% of participants. This finding is in line with the determined prevalence in a US study, which reported that 22.2% of the US non-institutionalized civilian population aged ≥18 years were disabled, but most of them had a low level of dependence [4]. There is, however, a slight discrepancy between the definitions of disability and dependency in daily living, so our results may not be directly comparable to the aforementioned study; however, we could not find any other research on the topic in the adult population which did not focus only on the elderly.

Our study focused on factors that are associated with the presence of dependence (and not its magnitude), as identification of people who might be dependent is of the highest importance to primary care teams.

We confirmed an association between dependence in daily activities and increasing age of patients over 60 years, which has also been found in some other studies [4, 5]. We think that there was no association between age and dependence in younger patients because the occurrence of dependence in this group is probably mainly associated with diseases or accidents.

We also found that dependence in daily activities was associated with the presence of at least one chronic disease [28, 32, 33]. However, the number of chronic diseases did not increase the probability of the dependency. In a study about foregone health care, Litwin and

**Table 3. Multivariate logistic model for dependence in daily activities (dependent vs. independent).**

| Variable | OR | 95% CI | *p* value |
|---|---|---|---|
| **Age** | | | <0.001 |
| Linear term | 0.93 | [0.85, 1.03] | 0.175 |
| Quadratic term | 1.00 | [1.00, 1.00] | 0.013 |
| **Gender** (female vs. male) | 0.74 | [0.50, 1.08] | 0.119 |
| **BMI** | | | 0.367 |
| Underweight vs. normal | 1.84 | [0.79, 4.27] | 0.159 |
| Overweight vs. normal | 1.11 | [0.71, 1.73] | 0.647 |
| Obese vs. normal | 0.88 | [0.54, 1.43] | 0.603 |
| **Family function** | 1.08 | [0.65, 1.80] | 0.765 |
| **Chronic diseases** | | | <0.001 |
| 1 disease vs. no disease | 2.13 | [1.30, 3.47] | 0.003 |
| 2 diseases vs. 1 disease | 1.43 | [0.89, 2.32] | 0.143 |
| 3 or more diseases vs. 2 diseases | 1.31 | [0.74, 2.31] | 0.360 |
| **Risk of falling** | 16.87 | [11.60, 24.55] | <0.001 |
| **Feelings of loneliness** | | | 0.001 |
| Linear term | 1.77 | [1.32, 2.38] | <0.001 |
| Quadratic term | 0.94 | [0.91, 0.97] | <0.001 |
| **Self-assessment of current health** | | | <0.001 |
| Linear term | 1.73 | [1.13, 2.66] | 0.012 |
| Quadratic term | 0.94 | [0.90, 0.97] | 0.001 |
| **Pain intensity** | 1.06 | [0.98, 1.16] | 0.158 |

Sample size n = 1814, AUC = 0.955 (bias-corrected 0.951), pseudo $R^2$ = 70.6% (bias-corrected 69.0%), likelihood ratio test *p*<0.001 (AUC = area under the ROC (receiver operating characteristic) curve, OR = odds ratio, CI = confidence interval, BMI = body mass index).

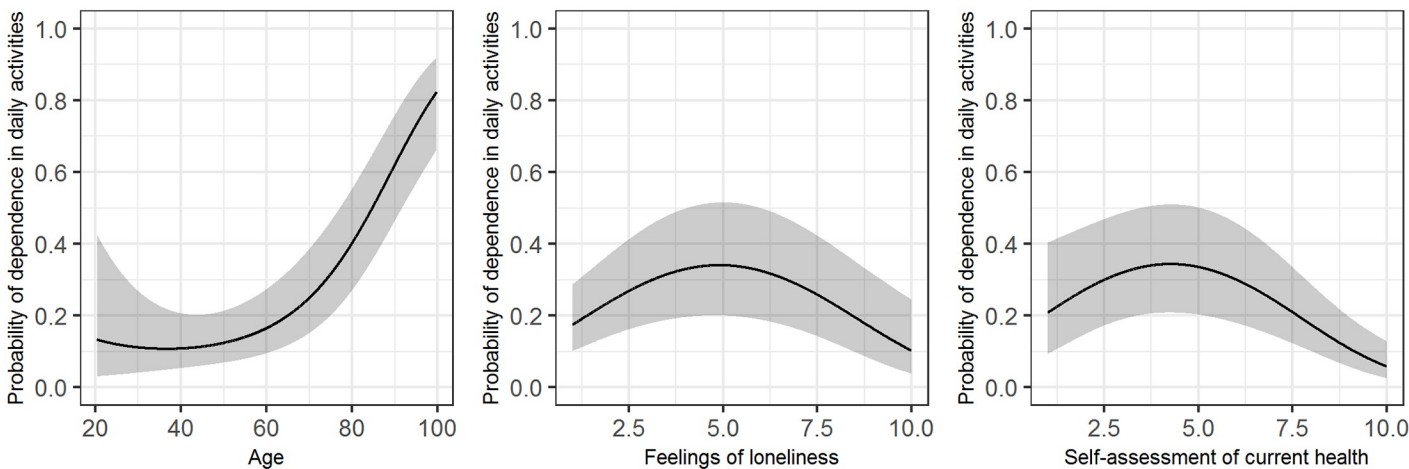

**Fig 2. Predicted probability of dependence in daily activities (black line) with 95% confidence intervals (grey area) in the multivariate logistic model (Table 3).** The figure shows the quadratic effects of age (left panel), feelings of loneliness (middle panel), and self-assessment of current health (right panel), where the other variables in the model are fixed to values indicating a healthy individual according to the variables in the model, except risk of falling (set to increased), gender to female, and age to its median of 61.5.

Sapir also did not find that the number of chronic diseases was important [16]. On the other hand, Laan et al. [34] found that the number of chronic diseases could be important; however, their population was limited to the elderly. Since we focused on all non-attenders regardless of their age, this may be a new finding, adding to the knowledge of that subgroup of primary care patients. For such patients, even only one chronic disease increases the odds of dependence in daily activities significantly (OR 2.1, 95% CI [1.3, 3.5], $p = 0.003$), which is certainly an unmet need to be managed by the primary healthcare team.

Some other characteristics of patients' clinical status (self-assessment of health, and risk of falling) were also shown to be important. Self-assessment of current health did not appear to be important for predicting dependence in daily activities among people with a low self-assessment of health, but it decreased the probability of dependence in those with a high self-assessment. Self-rated health has been found to be a good indicator of a health condition [35], and the ability to perform daily activities could be related to self-rated health [36]; these findings were confirmed by our study, but also revealed that different relations could exist for people with lower self-assessment of health vs. people with high self-assessment of health. This should be investigated further.

Risk of falling was positively associated with the presence of dependence in daily activities. Previous studies have already demonstrated this finding in populations older than 65 years [37, 38], but our study also confirmed this finding in the general population of non-attenders. This could be another unmet need of non-attenders. Since our study also included younger patients, it is therefore necessary to screen these kind of patients for risk of falling, thus preventing dependence in daily activities.

In addition, social characteristics (in our case feelings of loneliness) seem to be important in explaining dependence in daily activities. This finding was expected, as low social participation was found to be a protective factor against dependence [39], and loneliness was shown to be a strong factor independently associated with a functional decline [28]; functional decline can significantly affect dependence [5]. According to the results of our study, feelings of loneliness are another unmet need of the non-attenders. Only extremely lonely patients in our study were at lower risk of dependence; we speculate that these are patients that are living alone with no or very little help from others and are therefore independent.

Non-attenders are a specific group of individuals with significantly different characteristics from attendees. They may be subject to unequal health care [17, 40], but also may have unmet needs that are not being appropriately dealt with.

## Strengths and limitations

The strengths of our study were the large number of participants and high response rate. Moreover, all the health personnel who collected the data went through a unified education, enabling more reliable data collection.

There were several potential limitations to our study. We used our tool with eight different categories and calculated a total score [41, 42]. Another limitation is the cross-sectional design, which does not enable the detection of causal relations between variables. Furthermore, we included only participants from one region of Slovenia (out of ten), which does not permit us to generalize our results to the entire Slovenian population. However, we were able to include more than 80% of the non-attenders from this region, which gives us confidence that the results can be generalized at least for this region. An additional limitation could be that the presence of chronic diseases was self-reported. On the one hand, this could lead to a too low prevalence of some diseases, but on the other, some patients could have reported the presence of a chronic disease that actually has not yet been diagnosed. Studies show that self-reporting

of chronic diseases when compared to register data is in general reliable [43], but could differ according to different diseases (e.g. depression could be reported with low reliability [44]).

## Conclusion

A considerable proportion of family practice non-attenders have a mild dependence in daily activities. These people could benefit the most from interventions that could delay the onset and progression of disability. Integrated and coordinated care between different professional members of a family medicine team may help in lowering the number of patients that do not come to the practice regularly, and increase the recognition of dependent patients. Several unmet needs of the non-attender population were found in this study. Specific interventions for non-attenders could be implemented in practice. These could be the inclusion of community nurses in the active health care of non-attenders, screening for risk of falling and loneliness, and inclusion of new technologies for management and consultations. Such interventions could improve the non-attenders' health status and lower disability, consequently lowering the associated health-care costs. Moreover, the lay population and volunteers could be included in patients' long-term care in their homes to make them more efficient and less dependent in daily activities. Future studies are needed to explore the causes for foregone care in primary care population, to confirm or find new unmet needs, and to test the interventions for dealing with unmet needs.

## Supporting information

**S1 Table. Comparison of the outcome (dependence) and demographic characteristics between the subsample used in the multivariate model for dependence in daily activities (n = 1814) and the subsample that was omitted from the model on account of missing values for the covariates in the model (n = 185).** Note that there were no missing values with regard to the outcome or demographic characteristics.
(DOCX)

## Author Contributions

**Conceptualization:** Antonija Poplas Susič.

**Data curation:** Rok Blagus, Nina Ružić Gorenjec.

**Formal analysis:** Rok Blagus, Nina Ružić Gorenjec.

**Funding acquisition:** Antonija Poplas Susič.

**Methodology:** Antonija Poplas Susič, Zalika Klemenc-Ketiš, Rok Blagus, Nina Ružić Gorenjec.

**Supervision:** Antonija Poplas Susič, Zalika Klemenc-Ketiš.

**Validation:** Zalika Klemenc-Ketiš, Rok Blagus, Nina Ružić Gorenjec.

**Visualization:** Rok Blagus, Nina Ružić Gorenjec.

**Writing – original draft:** Antonija Poplas Susič.

**Writing – review & editing:** Zalika Klemenc-Ketiš, Rok Blagus, Nina Ružić Gorenjec.

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
