## [Decision Letter · Decision Letter 0]

21 Aug 2020

PONE-D-20-08631

Factors that determine dependence in daily activities: a cross-sectional study of family practice non-attenders from Slovenia

PLOS ONE

Dear Dr. Klemenc-Ketis,

Thank you for submitting your manuscript to PLOS ONE. After careful consideration, we feel that it has merit but does not fully meet PLOS ONE’s publication criteria as it currently stands. Therefore, we invite you to submit a revised version of the manuscript that addresses the points raised during the review process.

I apologize for the long throughput time of your original manuscript as I had to invite more than 40 researchers to eventually find two reviewers.

We look forward to receiving your revised manuscript.

Kind regards,

Kees Ahaus

Academic Editor

PLOS ONE

Journal Requirements:

3.We note that you have indicated that data from this study are available upon request. PLOS only allows data to be available upon request if there are legal or ethical restrictions on sharing data publicly. For information on unacceptable data access restrictions, please see http://journals.plos.org/plosone/s/data-availability#loc-unacceptable-data-access-restrictions.

Additional Editor Comments (if provided):

I have read this well-written paper with great interest. The study is based on a large sample and was designed in a way that a high response rate could be realized.

I think that in a revised paper the authors should pay more attention in the introduction to what the contribution of the study actually is. What was the gap in literature? What is added to literature, is it particularly interesting that all ages were in the sample, or is it interesting to study particularly non-attendees who may have unmet healthcare needs?

Please note, that reviewer 1 had concerns about the statistical analysis. She also brought forward important issues, such as the difference in the nature of disability for younger and older people.

In addition, I agree with her that it should be clarified why looking for interactions was beneficial, and what (based on argumentation) we can learn from interactions with dependence (such as, BMI x family functioning) or level of dependency (such as, family functioning x risk of falling). Finally, I suggest to improve the discussion which now reads as a summary of findings. The study definitely reveals interesting findings (e.g., dependence being associated with the presence of one chronic disease but not with multi-morbidity; a decrease of level of dependency with age within the group of dependent persons), but what can we learn from this study compared with existing literature, and what are implications?

Reviewers' comments:

Reviewer's Responses to Questions

**Comments to the Author**

1. Is the manuscript technically sound, and do the data support the conclusions?

Reviewer #1: Partly

Reviewer #2: Yes

2. Has the statistical analysis been performed appropriately and rigorously? 

Reviewer #1: No

Reviewer #2: Yes

3. Have the authors made all data underlying the findings in their manuscript fully available?

Reviewer #1: Yes

Reviewer #2: Yes

4. Is the manuscript presented in an intelligible fashion and written in standard English?

Reviewer #1: Yes

Reviewer #2: Yes

5. Review Comments to the Author

Reviewer #1: The interest of this study lies in its addressing a group that is important from a health care perspective: non-attenders (>5 years) of family practice. Also, the study includes a broad set of potential correlates of dependence, and has a very high response rate. Despite these positive points, I have a number of conceptual and methodological concerns that lowered my enthusiasm for this study.

Conceptually, the authors distinguish two outcomes: One dichotomous ‘non-dependent’ versus ‘dependent’, and one continuous ‘level of dependence’ – the latter only in the participant with at least some dependence. These are two conceptually different outcomes, and I would have liked to see a conceptual argumentation why the authors address both. In the Discussion, no distinction between the two is made whatsoever, which is one of the reasons why the Discussion is rather superficial.

The distinct concepts of dependence should be directly related to what the authors want to accomplish with their study – and this seems to be: to provide insights that enable specific interventions in the non-attendees who may have unmet health care needs. I would say that in this context, the most important outcome is the dichotomous one. In addition, it is of interest to show that most of the ‘dependent’ non-attendees are mildly dependent (but I miss a report of the distribution of the level of dependence within the dependent participants). However, personally, I cannot think of what purpose would be served by the fine-tuning of correlates, i.e. of level of dependence, beyond the correlates of dependence.

A second conceptual issue is the mixing of all ages of 18 and over. The nature of disability is quite different in younger and older people, even to the extent that there are two very distinct scientific disciplines that address these. In younger people, disability is often life-long from birth onwards or has an early onset. Most older people, in contrast, have led healthy independent lives and experience disability onset later in life. The authors recommend that there should be more studies on disability in younger people. However, these exist, but not in gerontology or geriatrics. I would suggest that the authors make use of this important conceptual age-distinction in interpreting their findings.

A third conceptual issue is application of interaction terms in the analytic models. Why do the authors look for interactions? What is the conceptual argumentation for each interaction? Now, their application of interaction terms seems ‘automatic’. If there is a good argumentation, moreover, each interaction term should be dealt with separately, and interpreted with care. As it is, the Discussion basically repeats what has been found, without giving sufficient thought to each interaction finding. For example, what to make of the interaction of BMI and family function? The picture shows that in underweight non-attendees with high family function, dependence is greater. What is cause and what is effect? It could well be that in some non-attendees, families are closer because their family member is sick – and thus underweight as well as dependent.

Here, the limitations of the cross-sectional design show up clearly. The authors use the word ‘determinants’ or ‘explanatory variables’, which suggests one causal direction, but it would be better to use ‘correlates’ instead, as this is not as suggestive of causality.

In the Statistical Analysis section, the authors state that they just study nine variables as correlates – “based on background knowledge”. What is this “background knowledge”? The argumentation should be place in the Introduction, including references.

Regarding the methodology, the data are generally well-described. In Tables 1 and 2, it can be seen that several characteristics have many missing values, the most for blood oxygen saturation. Apparently, and I am guessing here, these were measured rather than self-reported. In the Measures section, it should be described what is self-reported and what is measured by the nurse. In particular, were height and weight measured or self-reported? This variable was included among the nine correlates selected, so it is important to describe this correctly. As for the other variables, why are they included in Tables 1 and 2, as they are not analysed further?

In the analysis in the dependent sample, the number of younger people must be very small, because (Table 2) the lower quartile limit is at age 70. Thus, just over 100 participants are younger than 70.

In the Discussion, the authors describe the self-reported nature of the chronic diseases as a limitation. There is a number of studies that compare self-reported chronic diseases with register data. Such comparison would be all the more important in non-attendees of family practice, because usually a doctor has told the participant that they have a disease – which presupposes contact with a doctor. Thus, this limitation could be described more extensively.

In the Results, it seems that the authors derive main effects from multivariate models including interaction terms of these main effects. Main effects can only be derived from models without interaction terms that include one or two of the main effects. Furthermore, the AUC of the model in Table 3 is very very high. This may be due to the redundancy introduced by the inclusion of several interaction terms, some including the same variable.

Finally, the authors cite very little pertinent literature about foregoing health care – only one study in people who do not participate in health checks. This is not the same as not attending regular health care such as family physicians. However, there are numerous studies about foregoing health care, for example Litwin & Sapir, European Journal of Ageing 2009.

In conclusion, I would recommend to revise this manuscript, with less findings presented, but for each analysis a proper argumentation.

Reviewer #2: This manuscript represents contribution to better understanding factors that determine dependence in daily activities of family practice non-attenders and investigates dependence in daily activities of family practice non-attenders on the entire population of adults, not only on the elderly. The study presents the results of original research and presented results have not been published elsewhere.

In section Materials and methods authors clearly describe the type of study and settings, participants, outcome measures and statistical analysis.

In section Results authors clearly describe the sample (1999 patients) which represents the remarkable size and representativeness of the sample.

In section Discussion the authors logically link the results of this study to the results of previous and underline that this research is not focused only on the elderly which I consider as a major scientific contribution.

Conclusions are presented in an appropriate fashion and are fully supported by the data.

The article is presented in an intelligible fashion and is written in standard English, the research meets all applicable standards for the ethics of experimentation and research integrity and the article adheres to appropriate reporting community standards for data availability.

Minor Revisions

1. In the section Introduction, I suggest to add term patient safety

2. In the section Introduction, it is necessary to describe the importance of an interdisciplinary approach in identifying dependence in daily activities

3. In the section Discussion, it is necessary to discuss more about the fact that younger patients have higher level of dependence and suggest to family physicians to keep that fact in mind in their daily practicing

6. PLOS authors have the option to publish the peer review history of their article (what does this mean?). If published, this will include your full peer review and any attached files.

Reviewer #1: No

Reviewer #2: No

---

## [Author Response · Author response to Decision Letter 0]

5 Oct 2020

Dear Editor and Reviewers,

Thank you very much for giving us a chance to revise the article. We took into account all suggestions, performed a new analysis, and substantially rewrote the article.

We hope that we were able to sufficiently answer all comments and that now the article is clearer and could be considered for a publication in your journal.

Sincerely,

Zalika Klemenc Ketiš, on behalf of all authors

Additional Editor Comments (if provided):

I have read this well-written paper with great interest. The study is based on a large sample and was designed in a way that a high response rate could be realized.

I think that in a revised paper the authors should pay more attention in the introduction to what the contribution of the study actually is. What was the gap in literature? What is added to literature, is it particularly interesting that all ages were in the sample, or is it interesting to study particularly non-attendees who may have unmet healthcare needs?

Authors’ response: We added to the Introduction the text and references about foregone care, we also added what was the final purpose of the study. We hope that now we pointed out that our aim was to study particularly non-attendees and foregone care.

Please note, that reviewer 1 had concerns about the statistical analysis. She also brought forward important issues, such as the difference in the nature of disability for younger and older people.

In addition, I agree with her that it should be clarified why looking for interactions was beneficial, and what (based on argumentation) we can learn from interactions with dependence (such as, BMI x family functioning) or level of dependency (such as, family functioning x risk of falling). Finally, I suggest to improve the discussion which now reads as a summary of findings. The study definitely reveals interesting findings (e.g., dependence being associated with the presence of one chronic disease but not with multi-morbidity; a decrease of level of dependency with age within the group of dependent persons), but what can we learn from this study compared with existing literature, and what are implications?

Authors’ response: We revised the article according to all reviewers’ comments, which are also reflected in Editor’s comments. Please, see answers to reviewers (particularly Reviewer 1). We particularly revised the Discussion, to comply with the suggestions.

Review Comments to the Author

Reviewer #1: The interest of this study lies in its addressing a group that is important from a health care perspective: non-attenders (>5 years) of family practice. Also, the study includes a broad set of potential correlates of dependence, and has a very high response rate. Despite these positive points, I have a number of conceptual and methodological concerns that lowered my enthusiasm for this study.

Conceptually, the authors distinguish two outcomes: One dichotomous ‘non-dependent’ versus ‘dependent’, and one continuous ‘level of dependence’ – the latter only in the participant with at least some dependence. These are two conceptually different outcomes, and I would have liked to see a conceptual argumentation why the authors address both. In the Discussion, no distinction between the two is made whatsoever, which is one of the reasons why the Discussion is rather superficial.

Authors’ response: Based on all comments of the Editor and both Reviewers, we decided to present only the analysis based on a dichotomous outcome (non-dependent vs. dependent) as we agree that this one is the truly important one (mainly from clinical point of view). Furthermore, we think this simplifies the text and enables us to focus on few important results which we want to convey to the readers. We explained in the Discussion why we performed the multivariate analysis only for dichotomous variable.

The distinct concepts of dependence should be directly related to what the authors want to accomplish with their study – and this seems to be: to provide insights that enable specific interventions in the non-attendees who may have unmet health care needs. 

Authors’ response: Thank you, we added your suggestion to the Introduction as an aim of the study.

I would say that in this context, the most important outcome is the dichotomous one. In addition, it is of interest to show that most of the ‘dependent’ non-attendees are mildly dependent (but I miss a report of the distribution of the level of dependence within the dependent participants). However, personally, I cannot think of what purpose would be served by the fine-tuning of correlates, i.e. of level of dependence, beyond the correlates of dependence.

Authors’ response: We performed the analysis again to focus only on the dichotomous variable and to omit interactions (see also the comment below). We excluded the multivariate model for the level of dependence as we agree that the model is not important. Instead, we added the graph presenting levels of dependence as suggested. 

A second conceptual issue is the mixing of all ages of 18 and over. The nature of disability is quite different in younger and older people, even to the extent that there are two very distinct scientific disciplines that address these. In younger people, disability is often life-long from birth onwards or has an early onset. Most older people, in contrast, have led healthy independent lives and experience disability onset later in life. The authors recommend that there should be more studies on disability in younger people. However, these exist, but not in gerontology or geriatrics. I would suggest that the authors make use of this important conceptual age-distinction in interpreting their findings.

Authors’ response: As we rewrote the article also in line with other comments, we omitted the model for the level of dependence and with that a part of the discussion regarding younger patients, see also the comment below.. However, as the focus of our study are non-attendees, we analyzed the sample as a whole.

A third conceptual issue is application of interaction terms in the analytic models. Why do the authors look for interactions? What is the conceptual argumentation for each interaction? Now, their application of interaction terms seems ‘automatic’. If there is a good argumentation, moreover, each interaction term should be dealt with separately, and interpreted with care. As it is, the Discussion basically repeats what has been found, without giving sufficient thought to each interaction finding. For example, what to make of the interaction of BMI and family function? The picture shows that in underweight non-attendees with high family function, dependence is greater. What is cause and what is effect? It could well be that in some non-attendees, families are closer because their family member is sick – and thus underweight as well as dependent.

Authors’ response: Thank you for these important and very useful comments which made us think again about our models and what we wanted to present to the readers. We agree that the interactions were added automatically, without proper conceptual argumentations. Therefore, we omitted all the interactions, which gives a clear interpretation of the main effects. The new multivariate model for dependence without interactions still discriminates the data very well with AUC of 0.955 and bias-corrected AUC of 0.951. We rewrote the discussion with the aim to deal with each main effect separately and to also interpret the findings in the light of unmet needs of non-attenders and in the light of practical implication of the results.

Here, the limitations of the cross-sectional design show up clearly. The authors use the word ‘determinants’ or ‘explanatory variables’, which suggests one causal direction, but it would be better to use ‘correlates’ instead, as this is not as suggestive of causality.

Authors’ response: Thank you, we corrected the term to covariates.

In the Statistical Analysis section, the authors state that they just study nine variables as correlates – “based on background knowledge”. What is this “background knowledge”? The argumentation should be place in the Introduction, including references.

Authors’ response: We explained in the Method section why some covariates were omitted from the multivariate analysis and also added the appropriate references.

Regarding the methodology, the data are generally well-described. In Tables 1 and 2, it can be seen that several characteristics have many missing values, the most for blood oxygen saturation. Apparently, and I am guessing here, these were measured rather than self-reported. In the Measures section, it should be described what is self-reported and what is measured by the nurse. In particular, were height and weight measured or self-reported? This variable was included among the nine correlates selected, so it is important to describe this correctly. 

Authors’ response: For all variables, we described how they were collected, see Methods section.

As for the other variables, why are they included in Tables 1 and 2, as they are not analysed further. 

Authors’ response: We explained in the Method section why some covariates were omitted from the multivariate analysis and also added the appropriate references. However, we think it is important to describe the sample with as many variables as possible so the reader could get an objective insight into the characteristics of the sample. We added this sentence to the Methods section: Sample was thoroughly described using all available variables (Tables 1 and 2) that were measured in the larger study context (see ‘Type of study and settings’).

In the analysis in the dependent sample, the number of younger people must be very small, because (Table 2) the lower quartile limit is at age 70. Thus, just over 100 participants are younger than 70.

Authors’ response: Yes, that is true. As we omitted the model for the level of dependence in the dependent sample, we omitted also the part of the discussion regarding younger patients, so this is not an issue anymore.

In the Discussion, the authors describe the self-reported nature of the chronic diseases as a limitation. There is a number of studies that compare self-reported chronic diseases with register data. Such comparison would be all the more important in non-attendees of family practice, because usually a doctor has told the participant that they have a disease – which presupposes contact with a doctor. Thus, this limitation could be described more extensively.

Authors’ response: We described this limitation more extensively and added the appropriate references.

In the Results, it seems that the authors derive main effects from multivariate models including interaction terms of these main effects. Main effects can only be derived from models without interaction terms that include one or two of the main effects. Furthermore, the AUC of the model in Table 3 is very very high. This may be due to the redundancy introduced by the inclusion of several interaction terms, some including the same variable.

Authors’ response: Please see also the comment above about omitting interactions from the model. As we now omit all the interactions, the model gives a clear interpretation of the main effects. Furthermore, the AUC is still high (0.955). Moreover, we computed the bias-corrected AUC which is also high (0.951).

Finally, the authors cite very little pertinent literature about foregoing health care – only one study in people who do not participate in health checks. This is not the same as not attending regular health care such as family physicians. However, there are numerous studies about foregoing health care, for example Litwin & Sapir, European Journal of Ageing 2009.

Authors’ response: Thank you. We added the references on that topic and also some text into the introduction and discussion.

In conclusion, I would recommend to revise this manuscript, with less findings presented, but for each analysis a proper argumentation.

Authors’ response: We agree with that comment so we revised the manuscript according to it (only one model with clear interpretation of the main effects, the discussion about each one). We hope it is clearer now and that the message it conveys is strong.

Reviewer #2: This manuscript represents contribution to better understanding factors that determine dependence in daily activities of family practice non-attenders and investigates dependence in daily activities of family practice non-attenders on the entire population of adults, not only on the elderly. The study presents the results of original research and presented results have not been published elsewhere.

In section Materials and methods authors clearly describe the type of study and settings, participants, outcome measures and statistical analysis.

In section Results authors clearly describe the sample (1999 patients) which represents the remarkable size and representativeness of the sample.

In section Discussion the authors logically link the results of this study to the results of previous and underline that this research is not focused only on the elderly which I consider as a major scientific contribution.

Conclusions are presented in an appropriate fashion and are fully supported by the data.

The article is presented in an intelligible fashion and is written in standard English, the research meets all applicable standards for the ethics of experimentation and research integrity and the article adheres to appropriate reporting community standards for data availability.

Authors’ response: Thank you very much for these positive comments.

Minor Revisions

1. In the section Introduction, I suggest to add term patient safety

Authors’ response: Added as requested to the Introduction.

2. In the section Introduction, it is necessary to describe the importance of an interdisciplinary approach in identifying dependence in daily activities

Authors’ response: We described the importance of an interdisciplinary approach in the Introduction.

3. In the section Discussion, it is necessary to discuss more about the fact that younger patients have higher level of dependence and suggest to family physicians to keep that fact in mind in their daily practicing.

Authors’ response: This is no longer present in the paper as we omitted the model for the level of dependence as suggested by Reviewer 1, please see the comments above. In short, we wanted to make the paper more clear and to the point. The model for the presence of dependence is the truly important one from interventional point of view, so we omitted the one for the level and instead we now present level of dependence graphically. Moreover, in the subsample of dependent patients, the number of younger people is small (around 100 dependent participants are younger than 70), so we did not want to put an emphasis on the conclusions about them.

---

## [Decision Letter · Decision Letter 1]

24 Nov 2020

PONE-D-20-08631R1

Factors that determine dependence in daily activities: a cross-sectional study of family practice non-attenders from Slovenia

PLOS ONE

Dear Dr. Klemenc-Ketis,

Thank you for submitting your manuscript to PLOS ONE. After careful consideration, we feel that it has merit but does not fully meet PLOS ONE’s publication criteria as it currently stands. Therefore, we invite you to submit a revised version of the manuscript that addresses the points raised during the review process. I ask specifically to take note of the suggestions of reviewer 1 who has paid a lot of attention in a close reading of the paper. Please, decide carefully which suggestions you want to include in the final version of the paper. After receiving of your second revision I will not involve the reviewers anymore, as they both in principle advised positively.

We look forward to receiving your revised manuscript.

Kind regards,

Kees Ahaus

Academic Editor

PLOS ONE

Reviewers' comments:

Reviewer's Responses to Questions

**Comments to the Author**

1. If the authors have adequately addressed your comments raised in a previous round of review and you feel that this manuscript is now acceptable for publication, you may indicate that here to bypass the “Comments to the Author” section, enter your conflict of interest statement in the “Confidential to Editor” section, and submit your "Accept" recommendation.

Reviewer #1: (No Response)

Reviewer #2: All comments have been addressed

2. Is the manuscript technically sound, and do the data support the conclusions?

Reviewer #1: Yes

Reviewer #2: Yes

3. Has the statistical analysis been performed appropriately and rigorously? 

Reviewer #1: Yes

Reviewer #2: Yes

4. Have the authors made all data underlying the findings in their manuscript fully available?

Reviewer #1: No

Reviewer #2: Yes

5. Is the manuscript presented in an intelligible fashion and written in standard English?

Reviewer #1: No

Reviewer #2: Yes

6. Review Comments to the Author

Reviewer #1: The authors have thoroughly revised their paper, and it has become much more straightforward. Now, it is the details that matter. I have some questions on clarification, and a series of suggestions and comments to improve readability.

Abstract:

- In the Objectives, the second sentence should read: “Our goal … patients and its severity level, and …”

- In the Conclusions, instead of “have low level of dependence”, I suggest to write: “are dependent but at a low level”

- The last sentence is difficult to grasp: your study does not examine “differences”! Please rephrase.

Introduction:

- First paragraph: For readability, better combine sentences 2 and 4, and end with sentence 3.

- Second paragraph: Indeed most studies focused on older adults, but you cite only one study ([6]). Here, I would expect a reference to one or more review papers. Note, that reference [9] seems to be about frailty, which is not dependence (cf. Fried et al, your reference [14]).

- Paragraph on forgone care: Please state the direction of the association for each of the associated factors that you list.

- Same: You describe family practice non-attenders as a specific group, citing a study in family practice that finds factors associated with non-attendance [18]. However, exactly the same is done in the studies you cite in the previous sentence, and thus I would suggest to combine the description of the results from these two studies.

Material and methods

- Participants: The second paragraph starts with “From this study”, but it is not clear if you mean your own or the larger study mentioned under “Type of study and settings”.

- Outcome measures: You describe a score of 1 on dependence as “independent”, and scores 2 and higher as “dependent”. Please also describe how you classify the scores between 1 and 2.

- Outcome measures, second paragraph: No variables are “presented above” so please delete these words.

- Other variables: Please provide more explanation of the concepts going into Family function, as the APGAR is not widely known. Also, please provide the ranges for the MUST and the MORSE.

- Statistical analysis: I still think that the arguments for including or excluding variables should not be placed in this analysis section, but preferably in the Introduction, just before the end, or else at the end of the Measures section. In your example of reason (1) to exclude variables, please state which one you chose to include: BMI or waist circumference.

- Furthermore, if you are convinced that the variables that are not included in your multivariate analyses, are important enough to be reported in a table, you should pay some attention to them in your text. And preferably, you should use this background information in your interpretation of the findings in the Discussion section.

Results:

- I do not understand why you present the descriptive data in two tables, both even with the same title; there are continuous variables in both, so this cannot be a reason to separate them. Age should be reported with the other demographic characteristics. Moreover, these are not only demographic or clinical characteristics, but also psychosocial characteristics.

- It turns out that only 1814 participants are left in your main analysis. This begs the question how the 185 participants with incomplete data compare to those with complete data in terms of demographics and dependence. Please report.

- In reporting your main findings, instead of p-values, please report the actual estimates OR’s and their CI’s in the text, at least for the significant associations. This way, the reader can more easily grasp the size of the estimated association as well as its uncertainty.

- The sentence about self-assessed current health is a bit awkward: instead of “important” I suggest that you write that up to score 5 there was no difference in risk of dependence. This sentence is repeated in the Discussion section.

- I like the two figures, and would recommend including them in the main text, and not as supplementary material.

Discussion:

- Second paragraph: Here you explain that you did not do a regression analysis for the dependence score. However, now you have no research question regarding this score (except for showing its distribution), and so the reader does not expect you to do such a regression analysis. Meanwhile, your statement about what primary care teams should do is in itself valuable, but is better placed in the Conclusion.

- When discussing the role of age, I would be interested to read your thoughts about why there is no increase in risk below the age of 60 in this sample of non-attendees.

- When discussing findings from the literature, you tend to phrase these as facts (e.g., “the ability .. is related to self-rated health” or “loneliness is a strong factor ..”). However, if you cite only one study, I suggest that you phrase such findings more carefully, e.g., “the ability .. has been found to be related ..”.

- I do not quite understand your argument about loneliness, the role of social participation, and the adoption of a certain lifestyle. Please clarify. I also do not understand the last sentences of this paragraph: what does the fact that your sample is community-living have to do with living alone?

- If I understand it correctly, the next paragraph, starting with “Non-attenders are a special group” is intended as a summary of the main findings. However, the text does not keep its focus on non-attenders, and new literature is cited. (What is meant by “health disparities in behavioral risk factors”? The new literature is better omitted.) Please rewrite this paragraph.

- I suggest to insert a new heading “Strengths and limitations”. I consider the high response as one of the strengths of this study, but you have not mentioned this.

- As your first limitation, you mention the variety of combinations of ADL-items used in different studies. However, your next comment, that there is a large interrelation (not: co-relation) between the items, makes this problem less urgent.

- What do you mean by “predictive limitations”?

Conclusion:

- It is not appropriate to write your recommendations as imperatives, e.g., “should”.

Language: The text would need to be corrected in several places, preferably by a native English speaker.

Reviewer #2: (No Response)

7. PLOS authors have the option to publish the peer review history of their article (what does this mean?). If published, this will include your full peer review and any attached files.

Reviewer #1: No

Reviewer #2: No

---

## [Author Response · Author response to Decision Letter 1]

24 Dec 2020

Dear Editor,

Thank you very much for giving us a chance to revise the article again. We carefully read all the reviewer’s comments and revised the article accordingly. Detailed responses can be found below.

We hope that we were able to sufficiently answer all comments and that now the article is clearer and could be accepted for a publication in your journal.

Sincerely,

Zalika Klemenc Ketiš, on behalf of all authors

Review Comments to the Author

Reviewer #1: The authors have thoroughly revised their paper, and it has become much more straightforward. Now, it is the details that matter. I have some questions on clarification, and a series of suggestions and comments to improve readability.

Abstract:

- In the Objectives, the second sentence should read: “Our goal … patients and its severity level, and …”

- In the Conclusions, instead of “have low level of dependence”, I suggest to write: “are dependent but at a low level”

- The last sentence is difficult to grasp: your study does not examine “differences”! Please rephrase.

Authors’ response: Thank you for these suggestions. We took all of them into account and revised the abstract accordingly.

Introduction:

- First paragraph: For readability, better combine sentences 2 and 4, and end with sentence 3.

Authors’ response: We corrected the text as suggested.

- Second paragraph: Indeed most studies focused on older adults, but you cite only one study ([6]). Here, I would expect a reference to one or more review papers. Note, that reference [9] seems to be about frailty, which is not dependence (cf. Fried et al, your reference [14]).

Authors’ response: We added more citations and removed the reference Fried et al. from this paragraph.

- Paragraph on forgone care: Please state the direction of the association for each of the associated factors that you list.

Authors’ response: We corrected the text as suggested.

- Same: You describe family practice non-attenders as a specific group, citing a study in family practice that finds factors associated with non-attendance [18]. However, exactly the same is done in the studies you cite in the previous sentence, and thus I would suggest to combine the description of the results from these two studies.

Authors’ response: We combined the results as suggested.

Material and methods

- Participants: The second paragraph starts with “From this study”, but it is not clear if you mean your own or the larger study mentioned under “Type of study and settings”.

Authors’ response: We corrected the text so it is more clear now. 

- Outcome measures: You describe a score of 1 on dependence as “independent”, and scores 2 and higher as “dependent”. Please also describe how you classify the scores between 1 and 2.

Authors’ response: Thank you for this comment, we now state the categorization of scores in more detail.

- Outcome measures, second paragraph: No variables are “presented above” so please delete these words.

Authors’ response: The words were deleted.

- Other variables: Please provide more explanation of the concepts going into Family function, as the APGAR is not widely known. Also, please provide the ranges for the MUST and the MORSE.

Authors’ response: We added the explanation regarding family function and the ranges for MUST and MORSE.

- Statistical analysis: I still think that the arguments for including or excluding variables should not be placed in this analysis section, but preferably in the Introduction, just before the end, or else at the end of the Measures section.

Authors’ response: We moved this paragraph at the end of the Measures section.

In your example of reason (1) to exclude variables, please state which one you chose to include: BMI or waist circumference.

Authors’ response: BMI was chosen over waist circumference. This sentence was added to the text.

- Furthermore, if you are convinced that the variables that are not included in your multivariate analyses, are important enough to be reported in a table, you should pay some attention to them in your text. And preferably, you should use this background information in your interpretation of the findings in the Discussion section.

Authors’ response: The variables that are not included in the multivariate model, are in our opinion necessary to be presented to fully describe the characteristics of the sample. A short sentence was added in the results to describe the sample in more detail. However since, these variables were not included in the model, they were not included in the discussion where only results for the multivariate model are discussed.

Results:

- I do not understand why you present the descriptive data in two tables, both even with the same title; there are continuous variables in both, so this cannot be a reason to separate them. Age should be reported with the other demographic characteristics. Moreover, these are not only demographic or clinical characteristics, but also psychosocial characteristics.

Authors’ response: we chose to present the descriptive data in two tables, one for numerical and one for categorical variables. A common table would be, in our opinion, too difficult (and long) to read. We added the expressions “categorical variables” and “numerical variables” to the title of appropriate table. We also added, as suggested, the expression “psychosocial characteristics”.

- It turns out that only 1814 participants are left in your main analysis. This begs the question how the 185 participants with incomplete data compare to those with complete data in terms of demographics and dependence. Please report.

Authors’ response: Thank you for this important note, we added the comparison of demographic characteristics and dependence for these two subsamples (1814 and 185 participants) in the Supplementary Table 1. Additionally, we state that they are comparable in the section Statistical analysis.

- In reporting your main findings, instead of p-values, please report the actual estimates OR’s and their CI’s in the text, at least for the significant associations. This way, the reader can more easily grasp the size of the estimated association as well as its uncertainty.

Authors’ response: We agree and we added the ORs and CIs to the p values wherever possible (it is not possible for variables that were included in the model with quadratic effect).

- The sentence about self-assessed current health is a bit awkward: instead of “important” I suggest that you write that up to score 5 there was no difference in risk of dependence. This sentence is repeated in the Discussion section.

Authors’ response: We amended the text as suggested.

- I like the two figures, and would recommend including them in the main text, and not as supplementary material.

Authors’ response: Yes, we agree and they are already a part of the main text (it is the journal requirement that their titles are a part of supporting information).

Discussion:

- Second paragraph: Here you explain that you did not do a regression analysis for the dependence score. However, now you have no research question regarding this score (except for showing its distribution), and so the reader does not expect you to do such a regression analysis. Meanwhile, your statement about what primary care teams should do is in itself valuable, but is better placed in the Conclusion.

Authors’ response: We adapted the text so that the latter statement remains but the discussion on the possibility of different regression analysis is omitted.

- When discussing the role of age, I would be interested to read your thoughts about why there is no increase in risk below the age of 60 in this sample of non-attendees.

Authors’ response: We added the explanation to the text of discussion.

- When discussing findings from the literature, you tend to phrase these as facts (e.g., “the ability .. is related to self-rated health” or “loneliness is a strong factor ..”). However, if you cite only one study, I suggest that you phrase such findings more carefully, e.g., “the ability .. has been found to be related ..”.

Authors’ response: We rephrased the text where appropriate.

- I do not quite understand your argument about loneliness, the role of social participation, and the adoption of a certain lifestyle. Please clarify. I also do not understand the last sentences of this paragraph: what does the fact that your sample is community-living have to do with living alone?

Authors’ response: Thank you, we agree that this was unclearly written. We hope that it is more clear now.

- If I understand it correctly, the next paragraph, starting with “Non-attenders are a special group” is intended as a summary of the main findings. However, the text does not keep its focus on non-attenders, and new literature is cited. (What is meant by “health disparities in behavioral risk factors”? The new literature is better omitted.) Please rewrite this paragraph.

Authors’ response: We rewrote this paragraph in light of the suggestions.

- I suggest to insert a new heading “Strengths and limitations”. I consider the high response as one of the strengths of this study, but you have not mentioned this.

Authors’ response: Thank you, we added a new subheading “Strengths and limitations” and also mentioned the high response rate as a strength. 

- As your first limitation, you mention the variety of combinations of ADL-items used in different studies. However, your next comment, that there is a large interrelation (not: co-relation) between the items, makes this problem less urgent.

Authors’ response: We deleted this limitation, as we agree with the comment completely.

- What do you mean by “predictive limitations”?

Authors’ response: We deleted this expression from the text and paraphrased it differently.

Conclusion:

- It is not appropriate to write your recommendations as imperatives, e.g., “should”.

Authors’ response: We rewrote the text so it does not include the imperatives.

Language: The text would need to be corrected in several places, preferably by a native English speaker.

Authors’ response: the text was edited for English and proofread by a professional native speaker.

---

## [Editor Report · Decision Letter 2]

2 Jan 2021

Factors that determine dependence in daily activities: a cross-sectional study of family practice non-attenders from Slovenia

PONE-D-20-08631R2

Dear Dr. Klemenc-Ketis,

We’re pleased to inform you that your manuscript has been judged scientifically suitable for publication and will be formally accepted for publication once it meets all outstanding technical requirements.

Kind regards,

Kees Ahaus

Academic Editor

PLOS ONE
---

## [Editor Report · Acceptance letter]

7 Jan 2021

PONE-D-20-08631R2 

Factors that determine dependence in daily activities: a cross-sectional study of family practice non-attenders from Slovenia 

Dear Dr. Klemenc-Ketiš:

I'm pleased to inform you that your manuscript has been deemed suitable for publication in PLOS ONE. Congratulations! Your manuscript is now with our production department. 

Kind regards, 

on behalf of

Dr. Kees Ahaus 

Academic Editor

PLOS ONE